# A Novel Breast Cancer Xenograft Model Using the Ostrich Chorioallantoic Membrane—A Proof of Concept

**DOI:** 10.3390/vetsci10050349

**Published:** 2023-05-12

**Authors:** Marta Pomraenke, Robert Bolney, Thomas Winkens, Olga Perkas, David Pretzel, Bernhard Theis, Julia Greiser, Martin Freesmeyer

**Affiliations:** 1In Ovo Imaging Working Group, Clinic of Nuclear Medicine, Jena University Hospital, 07747 Jena, Germany; 2Institute of Organic & Macromolecular Chemistry (IOMC), Friedrich Schiller University Jena, 07743 Jena, Germany; 3Jena Center for Soft Matter (JCSM), Friedrich Schiller University Jena, 07743 Jena, Germany; 4Section Pathology, Institute of Forensic Medicine, Jena University Hospital, 07747 Jena, Germany

**Keywords:** animal models, ostrich chorioallantoic membrane, CAM, MDA-MB-231 cell line, xenograft

## Abstract

**Simple Summary:**

Animal testing is an important method in medical research and the development of new drugs (pharmaceuticals). Classic animal models feature mice or rats, but whenever possible, these animals should only be used for scientific purposes when absolutely necessary. Using less-developed forms, such as embryos not capable of pain perception, is considered an approach to reduce and replace adult animal testing. The use of chicken embryos for cancer research is well-known and features the implantation of cancer cells in order to form tumors (xenografts) on embryonic membranes (CAM). The tumor can be investigated via imaging methods, for example, nuclear medical imaging. However, chicken embryos are small and, therefore, require dedicated small animal imaging systems, which are expensive and require trained personnel. Therefore, we have investigated whether large ostrich embryos also are capable of growing tumors on the CAM. The large size of ostrich embryos would allow the use of routine imaging devices used for examinations in humans. We implanted breast cancer cells on the ostrich embryo CAM and successfully observed tumor growth. We suggest that the ostrich embryo is a suitable model for xenograft tumor imaging and cancer-related pharmaceutical research. This needs to be elucidated in further studies.

**Abstract:**

The avian chorioallantoic membrane (CAM) assay has attracted scientific attention in cancer research as an alternative or complementary method for in vivo animal models. Here, we present a xenograft model based on the ostrich (*struthio camelus)* CAM assay for the first time. The engraftment of 2 × 10^6^ breast cancer carcinoma MDA-MB-231 cells successfully lead to tumor formation. Tumor growth monitoring was evaluated in eight fertilized eggs after xenotransplantation. Cancer cells were injected directly onto the CAM surface, close to a well-vascularized area. Histological analysis confirmed the epithelial origin of tumors. The CAM of ostrich embryos provides a large experimental surface for the xenograft, while the comparably long developmental period allows for a long experimental window for tumor growth and treatment. These advantages could make the ostrich CAM assay an attractive alternative to the well-established chick embryo model. Additionally, the large size of ostrich embryos compared to mice and rats could help overcome the limitations of small animal models. The suggested ostrich model is promising for future applications, for example, in radiopharmaceutical research, the size of the embryonal organs may compensate for the loss in image resolution caused by physical limitations in small animal positron emission tomography (PET) imaging.

## 1. Introduction

The avian chorioallantoic membrane (CAM) assay is one of the most promising alternatives to conventional animal studies using rodents or other mammals [1,2,3,4]. It is a simple, versatile, fast and relatively time- and cost-efficient model [4]. It has attracted considerable attention in the field of tumor biology as an alternative model in accordance with the 3Rs guidelines (replacement, reduction and refinement) [5]. These guidelines state that whenever possible, animals should be replaced by less sentient alternatives such as invertebrates or in vitro methods. According to EU and international legislation, the CAM model system does not qualify as animal research if experiments are stopped before hatching [6].

The CAM is the transparent outermost extraembryonic membrane. The location of the CAM underneath the eggshell enables easy accessibility and observation of xenografted cells. One function of the CAM is to serve as respiratory organ for the embryo. Gases are exchanged via the CAM and through pores in the eggshell [1,7]. Furthermore, the CAM is responsible for the transport of calcium from the eggshell into the embryo to start mineralization and for water, sodium and chloride electrolyte reabsorption from the allantoic cavity where urinary products are discharged. The CAM is a highly vascularized extraembryonic structure connected to the embryo circulation by two allantoic arteries and one allantoic vein, which are associated with lymphatic vessels [1,8]. The high density of blood vessels creates an ideal milieu for efficient tumor growth due to the ubiquitous supply of oxygen, nutrients and growth factors [9]. At the beginning of incubation, avian embryos represent a naturally immune-deficient host, since the development of the lymphoid system starts in the late stage of incubation [3]. In contrast to the rodent model, most xenografts on the CAM are not rejected [1,10,11,12]. Mostly, cancer cells or patient-derived tissues are transplanted onto the CAM, followed by the formation of tumors within 2–10 days [1,13,14]. Among avian CAM models, the CAM of the chicken embryo is the best studied [15,16,17,18,19,20,21]. There is already a first report on quail CAM [22]. However, to the best of our knowledge, no work on the ostrich CAM has been reported to date.

Ostrich eggs have been already used for preclinical imaging. Studies have reported on the feasibility and technical success of positron emission tomography/computed tomography (PET/CT) and computed tomography (CT). In ovo imaging using ostrich eggs has been already described as a good alternative concept not only to common animal testing using rats or mice but also to chicken eggs. Due to their large size of approximately 20 × 15 cm, there is no need for dedicated small animal imaging devices, thus bypassing costly investments [23,24,25].

We provide a description of an in ovo ostrich CAM model xenografted with human breast carcinoma cells. We utilized the highly invasive MDA-MB-231, a cell line that is negative for the estrogen receptor (ER-), HER2 receptor and progesteron receptor (PR), thereby serving as an in vitro model of triple-negative breast cancer. In this work, we present a proof of concept showing that the ostrich embryo CAM can be used for grafting tumor cell lines, with the MDA-MB-231 serving as a model cell line.

## 2. Materials and Methods

All in ovo experiments were in compliance with European law (Directive 2010/63/EU of the European Parliament and of the Council of 22 September 2010 on the protection of animals used for scientific purposes), which, from a regulatory point of view, does not consider CAM assays as animal experimentation [1]. The Office for Consumer Protection of Thuringia/Germany permitted the in ovo study protocol (registration number 22-2684-04-02-114/16) and confirmed the in ovo experiments not to be an animal study.

### 2.1. Ostrich Embryos

Fertilized ostrich eggs (*struthio camelus*) were purchased from a local farm (Silbertaler Straußenfarm, Bürgel, Germany) and incubated at 37.6 °C and 20% relative humidity in a specialized incubator (J. Hemel Brutgeräte GmbH Co., KG, Verl, Germany). All eggs were placed in an upright position in the incubation trays. The incubator shelves were set to alternate tilting of 45 degrees every 2 h for the whole incubation period to prevent embryonal structures sticking to the egg membrane, thus simulating natural breeding conditions. After incubation of the eggs for 20 days, they were analyzed for fertilization and embryonic development using a candling lamp.

### 2.2. Investigation of the Suitable Day for Cell Grafting

A total of eleven fertilized ostrich eggs was used to analyze the most suitable embryonic development day (DD) for the engraftment of the tumor cells. All eggshells were washed with a cotton swab soaked with 70 % ethanol solution and left to dry. The eggs were candled with a light-intensive candling lamp (Tempo, No. 119, Breker Ltd. & Co., KG, Ruethen, Germany) at the side of the eggs as well as in the air cell. This allowed a diaphanoscopy-like depiction through the eggshell, similar to published results [23]. In the air cell, a shell access circular window with a diameter of ~10 cm, was carefully opened using a rotating cutter (Dremel, 3000; Bosch Powertools B.V, Breda, Netherlands). Afterwards, the window was sealed with a transparent adhesive tape (Transpore^®^ 3M, Neuss, Germany) to prevent contaminations, and the eggs were returned to the incubator. This shell opening procedure was performed on different embryonic development days, starting from DD 1 up to DD 20. The eggs were checked every day for changes through visible observation using a candling lamp. The continuous gas exchange (carbon dioxide production) of regular embryonal development was controlled by the vitality measurement, similar to our experiment reported elsewhere [23,25]. The observation of the development of the CAM was stopped on DD 34.

### 2.3. Cell Culture Line and Culture Conditions

The human breast carcinoma cell line MDA-MB-231 was incubated in Dulbecco’s modified Eagle´s medium (DMEM, Capricorn Germany, Meuspath, Germany) supplemented with 10 % fetal bovine serum (FBS, Capricorn Scientific, Ebsdorfergrund, Germany) and penicillin/streptomycin (Pen/Strep, 100 IU/mL, Capricorn Scientific, Ebsdorfergrund, Germany). The culture was kept at 37 °C in a humidified atmosphere of 95% air and 5% CO_2_.

### 2.4. Cancer Cell Engraftment

We adopted a similar protocol to the chicken embryo CAM model with slight modifications [17,26]. Eight fertilized ostrich eggs were used for the cancer cell implantation. On DD 20, eggs were gently washed with 70% ethanol solution and a window of an approximate 10 cm diameter was drilled in the air cell. The window allowed the implantation of the cells. In the white inner eggshell membrane covering the CAM, a small incision (length of 1 mm) was very carefully made with a scalpel without damaging the CAM. The preparation of the cell suspension was performed on ice. A total of 1 × 10^6^ cells suspended in 100 µL of 70% Matrigel (Standard Matrigel Matrix, Corning, Berlin, Germany) were grafted onto the CAM of 2 eggs. In another cell grafting series, 2 × 10^6^ cells suspended in 50 µL of serum free-DMEM were mixed with 50 µL of Matrigel and applied to the CAM of five ostrich eggs. One ostrich egg was seeded with 6 × 10^6^ cells suspended in 250 µL of 60% Matrigel. The cell suspension was transferred directly onto the CAM accessible through the hole in the white inner eggshell membrane using a standard laboratory pipet. After cell engraftment, an adhesive tape was used to cover the window, and eggs were kept in the incubator for further growth. Tumor growth and embryo viability were examined daily until day DD 34, as was already reported [22,24].

### 2.5. Xenograft Harvesting and Histological Analysis

On DD 34, the ostrich embryos were euthanized with pentobarbital (1.0 mL, 1 mL/kg) which was injected i.v. via a blood vessel accessible via the additional square window (16 cm^2^) on the side of the egg. In the air cell, the white membrane covering the CAM was removed and the CAM areas with a tumor in the middle were carefully cut and fixed in formaldehyde (FA, 4%). After fixation, the tumor tissue was removed from FA and stored in ethanol, followed by paraffin embedding. Embedded tumors were sectioned at 3 µm (using a microtome) and baked for 20 min at 77°C, followed by deparaffinization with xylene and rehydration using an ethanol gradient (100%, 95%, 70%) then a distilled water wash. Hematoxylin and eosin (H&E) staining was performed for the histological analysis of the tumor tissues.

## 3. Results

### 3.1. CAM Development and Determination of the Optimal DD for Cell Grafting

To our knowledge, there is no literature on the CAM development of ostrich embryos yet. Therefore, we carried out most of the experiments based on well-established protocols suitable for the chicken embryo CAM [15,16,17,18,19,20,21]. The development of ostrich embryos (42 days) is twice as long as chicken embryos (21 days); therefore, some adaptations from the established chicken embryo protocols had to be made to suit the novel ostrich model. In particular, the identification of the optimal DD for cell grafting was a crucial point.

During embryo development, three membranes are formed: the amnion, the yolk sac membrane and the chorioallantoic membrane (CAM) [3]. In chicken embryo development, the CAM develops between days 3.5 and 10 through the fusion of the splanchnic mesodermal layer of the allantois and somatic mesoderm of the chorion [3,27]. As a first approximation for the ostrich embryo model, this timescale was multiplied by two as this is in agreement with the ratio of total developmental time between ostriches and chickens. Therefore, we assumed that the ostrich embryo CAM is formed between days 7 and20. Likewise, since in the chicken embryo CAM model, cancer cell engraftment is commonly performed between DD 7 and 10, for the ostrich embryo CAM model, DDs 14–20 could be the most promising for cell engraftment. We sought to verify this assumption through continuous observation of the CAM vessel development for almost the entire ostrich embryo developmental period.

Of the 11 embryos investigated for the vascularization of CAM monitoring, two embryos died by developmental day 21. A typical visual observation is shown in Figure 1 with a representative depiction of blood vessels on DD 20 (Figure 1a) and on DD 27 (Figure 1b). The earliest we observed the first visible vessels was on DD 10, which is in good agreement with the chicken embryo model, corresponding to DD 4 [7]. Usually, on DD 20, a rich vascular network was expanded (Figure 1a). Based on correlations with the chicken CAM development [27], we assumed the ostrich CAM development should be finished by DD 20, at the latest. Therefore, we oriented the cell engraftments on DD 20, which corresponds to DD 10 of chicken embryos, and decided to apply cancer cells on this day due to the high vascularity of the ostrich CAM membrane.

### 3.2. Tumor Cell Grafting and Tumor Growth Success

The general scheme followed in all experiments is presented in Figure 2. Our main aim was to test whether the ostrich embryo CAM is suitable for tumor formation. On DD 20, eight fertilized eggs were chosen for the grafting process.

Initially, we utilized 1 × 10^6^ carcinoma cells for engrafting onto the CAM of two ostrich embryos, since this cell number is commonly used in the in ovo chicken CAM model [17]. Usually, under these conditions, the MDA-MB-231 cell line injected onto the chicken CAM leads to tumor formation with a size of 5 mm over a 9-day period [17,26]. However, we did not observe tumor growth on the ostrich embryo CAM under these experimental conditions. It is known from the chicken CAM model that an MDA-MB-231 cell amount below 1 × 10^6^ does not lead to efficient tumor growth [17]. Therefore, we decided to increase the cancer cell amount to 2 × 10^6^ cells for the CAM engraftment of five more embryos. Under these conditions, in two cases, we observed a tumor growth on the CAM 14 days after engraftment (Figure 3). One of these observed tumors had a spherical form with a size of approximately 4 mm (Figure 3 a), and the tumor observed on the CAM of a second embryo was formed in an irregular shape with a length of 1.2 cm. For the other three embryos, no visible changes on the CAM were observed. This means that, statistically, 40 % of the ostrich embryos were able to form a tumor on the CAM. A further increase in the cancer cell amount to 6 × 10^6^ (one embryo) did not indicate successful tumor formation on the ostrich CAM either.

The histopathological analysis of both tumors using H&E staining revealed a diffuse proliferation of epithelial tumor cells within the mesenchymal stroma of the chorioallantoic membrane (Figure 4). The tumor cells showed marked cytonuclear atypia and a high proliferative activity with numerous, often atypical, mitotic figures (yellow arrows) (Figure 4a,b). Scattered inflammatory cells (green arrow) and nucleated erythrocytes (red arrow) were detectable between the neoplastic cells. In the background, there was a dense network of slit-like capillary vessels (blue arrow).

## 4. Discussion

We demonstrated that in principle, the ostrich embryo CAM system is capable of forming tumors of a breast cancer cell line, similar to the chicken CAM model. Contrastingly to the method established for the chicken CAM, we observed the best results when using 2 × 10^6^ cells instead of 1 × 10^6^ cells. However, our studies encompassed only a very limited number of subjects, so the results do not yet exhibit sufficient statistical certainty. In particular, a larger number of subjects would be required to verify the optimal cell number. Furthermore, although DD 20 was identified through visual observation to be a suitable time point for tumor cell engraftment, this should be further supported in the future by analysis of the embryo immune system at this developmental stage. Additionally, in an ongoing study, we will more closely monitor tumor growth, thereby determining whether tumors on the ostrich CAM have the potential to grow larger than on the chicken CAM.

The ostrich CAM model, due to its larger surface area and longer time for embryo development, opens a new way as an alternative to the chicken embryo CAM, allowing prolonged examination time and potentially even the application of multiple tumors per embryo. Thus, a wide range of new cancer therapies could be studied in preclinical experiments. Our results demonstrate the principle ability of the ostrich CAM system to form a tumor. However, reliable tumor growth could not yet be observed in all experiments, and further optimization of the protocol is needed.

The size of the ostrich embryo makes it particularly interesting for preclinical imaging in the field of nuclear medicine. Nuclear medical imaging employs radioactive substances to visualize molecular targets, such as tumor cell receptors. Common models used in small animal imaging are adult mice or rats, or chicken embryos. However, particularly in small animal positron emission tomography, spatial image resolution is limited by the physical positron range of the radionuclides, which lies in the range of a few millimeters. Therefore, small anatomical structures in these small animals may not be depicted with sufficient image quality. Contrastingly, the ostrich embryo model may compensate for the resolution loss caused by the positron range, simply due to its larger size. Additionally, dedicated small animal imaging devices are much less prevalent compared to clinical devices. As we have recently shown, the ostrich embryo model may be used for preclinical imaging studies using conventional clinical PET scanners; therefore, preclinical imaging with avian embryos is no longer limited exclusively to dedicated small animal facilities [25]. Thus, we identified the ostrich embryo as a suitable model for the development and investigation of TNBC-targeting PET radiotracers, which are currently being investigated in our facility.

Compared to chicken embryos, the availability of ostrich embryos is more limited as it relies on the existence of ostrich farms. However, since harvested eggs are robust enough to be shipped, ostrich farms do not necessarily have to be in close proximity to the research facility. Ostrich farming is a popular agricultural branch, especially in Germany and Africa, and steady egg supply can be ensured between April and October. While the size of the eggs requires some special equipment, such as an incubator of adequate dimensions, the embryo size brings some advantages regarding methodology. For example, blood vessels are much thicker in ostrich embryos compared to chicken embryos, making intravasal injection and blood sampling (e.g. for metabolite analysis) more feasible. To conclude, the data presented here support that the ostrich embryo model is suitable for implanting xenografts and, thus, provides an adequate alternative to established in ovo chicken models or small animal models, whenever parameters linked to the animal size (such as blood volume, organ diameter) are of particular importance.

## 5. Conclusions

The ostrich embryo CAM may be used for implantation of xenografts for preclinical studies, particularly when large objects are required.

## Figures and Tables

**Figure 1 vetsci-10-00349-f001:**
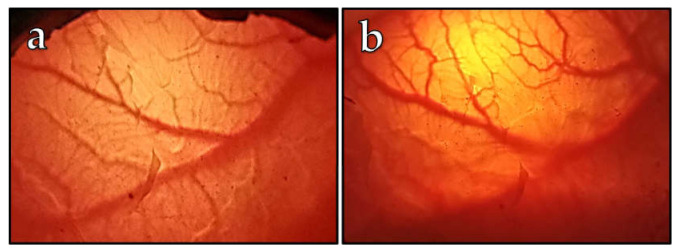
Visual observation of the blood vessel formation in the ostrich CAM on different embryonic development days: 20 (**a**) and 27 (**b**), performed with the use of a candling lamp placed on the side of the egg, close to the air cell. From DD 20 to DD 27, an increased visible vascularization of the CAM and a vascular diameter enlargement were observed.

**Figure 2 vetsci-10-00349-f002:**
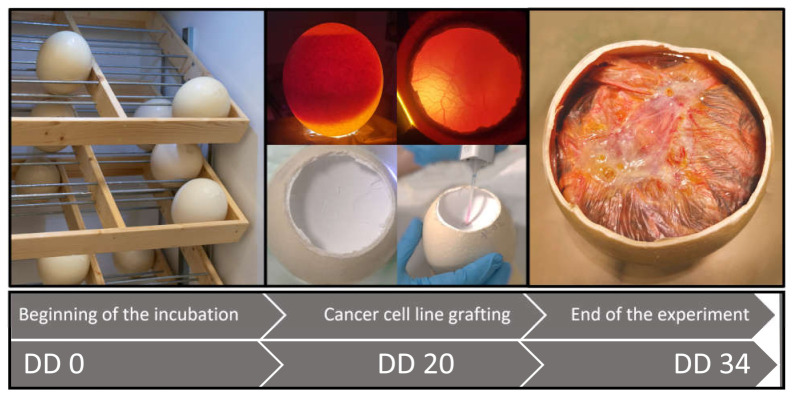
Procedure of the MDA-MB-23 cell line implantation onto the ostrich CAM. On DD 0, the egg incubation starts. Then, 20 days later, the fertilized eggs are selected for the experiments. The egg is opened in the air cell area and candled with a tube lamp, and the cells are grafted in the middle position onto the CAM. On DD 34, the embryos are euthanized and the sample for the histological examination is prepared.

**Figure 3 vetsci-10-00349-f003:**
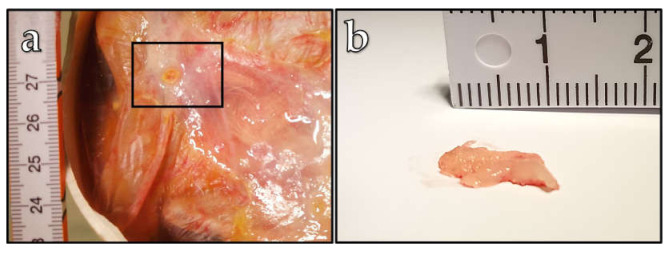
Typical grown tumors on the ostrich chorioallantoic membrane of two different embryos (**a**,**b**).

**Figure 4 vetsci-10-00349-f004:**
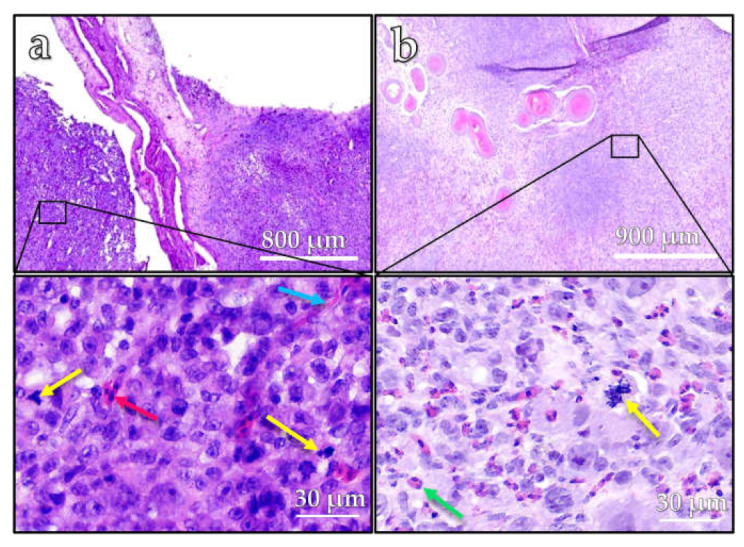
Overview of the ostrich CAM of two different ostrich embryos (**a**,**b**), in both cases showing a dense proliferation of epithelial tumor cells with cytonuclear atypia and a high proliferative activity with numerous, often atypical, mitotic figures (yellow arrows), scattered inflammatory cells (green arrow), nucleated erythrocytes (red arrow) and a dense network of slit-like capillary vessels in the background (blue arrow).

## Data Availability

The data presented in this study are available on request from the corresponding author. The data are not publicly available due to being in a non-standardized format.

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
