# Peer review of "A Novel Breast Cancer Xenograft Model Using the Ostrich Chorioallantoic Membrane—A Proof of Concept"

_vetsci, 2023, doi:10.3390/vetsci10050349_

Round 1

Reviewer 1 Report

In this paper, Pomraenke describe the in ovo ostrich CAM model xenotransplanted with human breast cancer cells and used the highly invasive triple negative breast cancer cell line MDA-MB-231.

It is a nice work that for the first time shows the application of the ostrich CAM as a model to study breast cancer. The size and development time of the ostrich and the large CAM area compared to CAM in the chicken makes this model interesting also for PET imaging.

Author Response

We thank the reviewer for his or her kind support and evaluation!

Reviewer 2 Report

The study by Dr. Marta Pomraenke and co-workers propones the use of ostrich chorio- allantoic membrane (CAM) to study breast cancer biology in vivo.

The CAM assay on avian embryos (such as quail, turkey and duck) is already well known and described in many publications, thus I do not think this study is very innovative.

Author Response

We agree that the CAM assay in principle is a well-established topic and transferring the method to the largest known bird embryo species (ostrich), just for the sake of pure methodology, would not be vastly innovative. However, we consider the ostrich embryo a particularly relevant model with future application in the field of radiotracer development and nuclear medical research, since the ostrich embryo is likely the only bird embryo large enough to be visualized on conventional and widely available clinical PET/CT scanners. We have added a paragraph into the discussion section of the revised version of the manuscript, going more into detail regarding the potential of the model in this particular field.

Reviewer 3 Report

This is a proof-of-concept study on feasability and practicability of an ostrich CAM model. This is an interesting contribution to the field.

Minor remarks:

- The authors could comment more specifically on the benefits of using ostrich instead of chicken eggs. What kind of experiments could be only carried ot with the ostrich model?

- Practicability seems rather low, due to availibility and also the sheer size of the model, which probably does not help when investigating purely cellular or molecular biological processes. Have attempts be made to use the ostrich model with methods such as intravital 2-photon-microscopy?

- What plans do the authors have for this model? What kind of experiments are planned beyond the pure methodolical advance?

Author Response

Minor remarks:

>> Practicability seems rather low, due to availibility and also the sheer size of the model, which probably does not help when investigating purely cellular or molecular biological processes. Have attempts be made to use the ostrich model with methods such as intravital 2-photon-microscopy?

We thank the reviewer for his or her insightful evaluation and comments that encouraged us to go more into detail in the revised version of the manuscript for the sake of providing more information regarding benefits, limits and potentials of the model.

We have provided an additional paragraph in the discussion section of the revised manuscript, reflecting on questions of availability of ostrich embryos and specific technical requirements, but also on the advantages that we see in regard to the embryo/egg size.

We plan to install a specifically designed stereo microscope suitable for imaging the cellular level. Also, we are currently establishing an ex-ovo approach, where the embryo is kept outside of shell, planning to make microscopy during embryo development feasible.

>> The authors could comment more specifically on the benefits of using ostrich instead of chicken eggs. What kind of experiments could be only carried ot with the ostrich model?

>> What plans do the authors have for this model? What kind of experiments are planned beyond the pure methodolical advance?

We have provided an additional paragraph at the end of the manuscript, going into detail on both, the advantages of the ostrich embryo model due to its size, particularly in the field of nuclear medical imaging research, and regarding our plans to develop TNBC specific radiotracers, using the presented ostrich in-ovo model.

Round 2

Reviewer 2 Report

The manuscrpit is now ready for pubblication.